# AI-based opportunistic analysis of the CT images during COVID (2021): Does living in a metropolitan area affect the vertebral body mineral density in older people?

## Abstract

The aim of the study is to reveal the problems of using information on several territorial units (districts) of an integral urban agglomeration for an identification of interspatial peculiarities of living in a particular area by estimating vertebral bone mineral density (BMI) in Moscow residents aged 50 and older. The results of this study will provide hypothesis testing for a pilot project to create a model of spatial exposure to the urban environment (Computer Vision Experiment; Clinical Trial: NCT04489992), which will form the basis of identifying individuals (groups of individuals) at risk of "accelerated" occupational aging and musculoskeletal diseases.

## 1 Introduction

```
Shifting from neoclassical economics to ecological economics
offers the opportunity to move to a system that is more focused on
the promotion of good health from the outset|a move to prevention
rather than cure.
``` Cited Deschênes & Greenstone (2011)

Existing open questions pose a number of pressing challenges for the organization of the healthcare system of Moscow, which relate to: (a) the application of existing results of other studies due to their contradictory heterogeneity obtained in high- and low-income countries Matsuzaki et al. (2015); ((b) the lack of local data on the residents of individual districts (municipal units) of the metropolitan area, taking into account individual spatial information, which is important for decision-making in the field of social and health care of the residents of a municipality. The aim of the study is to reveal the problems of using information on several territorial units (districts) of an integral urban agglomeration for: 1) identification of interspatial peculiarities of living in a particular area by estimating BMI in Moscow residents aged 50 and older; 2) clarification of existing local-spatial peculiarities of territories on the basis of comparison of BMI values for the American population (UCSF); 3) determining and discussing the influence of individual spatial exposure on the BMD trajectory of Moscow residents aged 50 and older. The results of this study will provide hypothesis testing for a pilot project to create a model of spatial exposure to the urban environment (Computer Vision Experiment; Clinical Trial: NCT04489992; `https://clinicaltrials.gov/ct2/show/NCT04489992`), which will form the basis of identifying individuals (groups of individuals) at risk of "accelerated" occupational aging (physical limitations and aptitude) and musculoskeletal diseases (for example, osteoporosis).

## 2 Methods

The aim of our study was to determine the differences in the distribution of BMD indicators (vertebral bodies) according to CT data of the chest organs using an AI algorithm in three spatially distributed observation groups.

## 2.1 Socio-environmental data and spatial impact on urban population

To verify the relevance of the problems initially analyzed statistical data on the urban health system based on geospatial division (by city district units) socio-economic and medical data (diseases of the Moscow population) Bocharova et al. (2021).

## 2.2 Pre-preprocessing AI-calculated CT data

BMD assessment was performed using an AI algorithm including pretrained neutral networks for anchor-free vertebra detection Zakharov et al. (2022). The mineral density was determined by measuring the X-ray density of the vertebral bodies Th11- L1 (HU) with AI-based on an opportunistic data-driven analysis of CT 3D images of the chest and performed in three polyclinics and two city hospitals. After that, the BMD values were determined by asynchronous QCT et al. (2016), taking into account calibration data obtained using a phantom with known normative BMD values Petraikin et al. (2019), which allowed us to obtain age dependences of the BMD distribution for men and women, which were then compared with the normative data Faulkner et al. (1993) (UCSF).

## 2.3 Analysis of older people in male and female groups

Patients with signs of COVID-19-associated viral pneumonia were referred for CT diagnosis during June 2021. The distribution of the total sample by severity (according to the Russian accepted CT1-4 classification) was CT1/CT2/CT3/CT4: 45.3/18.4/3.8/0.6 percent, respectively; no signs of viral pneumonia in 31.9 percent. Further, patients (men and women) aged 50 years and older, who were attached under the territorial principle under the MHI in three medical organizations of the city health care system, located (one) in the Central, (two) in the Southern administrative districts, were selected from the general sample, where they underwent the laboratory CT investigation. Patients with CT measurements of density in the vertebrae with a compression deformity of more than 25 percent were excluded from the sample. 1,135 women and 718 men were included in the analysis.

## 2.4 Data anonymization

For the purposes of subsequent processing and compliance with confidentiality conditions, ethical standards, in accordance with the Helsinki Declaration and respect for the rights of patients, the data were anonymized. A digital code was assigned to each data record that included a territorial attribute.

## 2.5 Statistical analysis

Statistical analysis was carried out using the IBM ® SPSS ® Statistics 20 statistical package (IBM, NY, USA). The patients with strict territorial attachment to Moscow city's polyclinics (three ones in the various spatial circumstances and distance connections of traffic ways, see Appendix - Supplementary Fig. 1) were selected from the target group for analysis taking into account gender, age and residence. All observation groups were compared in pairs. To confirm the heterogeneity of the three groups of each of the sexes, a test was carried out, Mann-Whitney and Kolmogorov-Smirnov nonparametric criteria were used to compare the ones.

## 3 Results

The age distribution of BMD values obtained from the results of AI-calculations in women is well comparable with the UCSF normative curve, despite the complex nonlinear nature of the dependence, where differences were unreliable for all age (50 yr. and older) intervals of the normative curve. In men, a significant underestimation of BMD values were revealed on average (z=-0.631) compared with the standard (UCSF) values for the age range of 50-75 years. For the most senior age groups (75+ years), the differences were unreliable. At the next stage, patients with strict territorial attachment to Moscow city's polyclinics (three ones in the various spatial circumstances and distance connections of traffic ways) were selected from the target group for analysis taking into account gender, age and residence. It was shown that there is a significant decrease in BMD values of locals from most remoted area from the city center to the south, both in men (p=0.04) and in women

(p=0.03). In addition, integral indicators of primary morbidity of the urban population older than working age for diseases of the musculoskeletal system and connective tissue were analyzed, which for residents of the central part (central administrative region of the city) were on average 109.25 higher in 2016-2020 (SD 0.023) compared with those for residents of the southern part; and integral indicators for dispensary patients – were also higher by 53.09 percent (SD 0.899), respectively Zlobina et al. (2015); Nikitina et al. (2019); Khramtsova et al. (2020).

## 4 CONCLUSION

The purpose of this study is to attract the attention and inform urban designers and politicians about sustainability and the need to analyze information about how specific areas of the city with the current environmental situation are related to detailed medical and social medical aspects of the etiology of health disorders.

AUTHOR CONTRIBUTIONS

Authors declare contributions: ...

ACKNOWLEDGEMENTS

The research was prepared ...

URM STATEMENT

The authors meets the URM criteria of ICLR 2023 Tiny Papers Track.

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

# A APPENDIX

Supplementary: Fig. 1 Supplementary: Table 1 [doi] `https://github.com/.../ICLR/BMDtable`

Supplementary: Highlights

## HIGHLIGHTS

It is necessary to highlight the conclusions that consist in the following provisions:

• AI algorithms, in particular pre-trained neutral networks for anchor-free vertebra detection (Zakharov et al., 2022), in addition to effectively solving clinical problems of radiation diagnostics, is also be a tool for performing urban studies;

• the BMD values of elderly residents of the two territories of the megalopolis Moscow (Russia) with a different structure of the urban environment significantly differ, and as well as they differ from the normative values for the American population (UCSF); and

• for the most adequate reflection of the level of risks of the occurrence and development of diseases, in particular of the musculoskeletal system in older people, it is necessary to take into account the area of city residence, spatial circumstances and distance connections of traffic ways, other related environmental factors to assess and plan morbidity rates in the public health system.

