# OpenReview forum: "AI-based opportunistic analysis of the CT images during COVID (2021): Does living in a metropolitan area affect the vertebral body mineral density in older people?"
_ICLR.cc/2023/TinyPapers — Submitted to Tiny Papers @ ICLR 2023_

### Official Review · Reviewer_PH5h · 2023-04-01

**Confidence:** 4

**Summary Of Contributions:**

The paper aims to identify interspatial peculiarities of living in different areas of Moscow by estimating vertebral bone mineral density (BMD) in residents aged 50 and older. The study uses information on several territorial units (districts) of an integral urban agglomeration to reveal the problems of using such information for decision-making in the field of social and health care of the residents of a municipality.

**Rating:**

Great Start (GS): a submission which meets some of the reviewing criteria but has room for improvement

**Strengths And Weaknesses:**

Strengths:
* The paper clearly states the background information about the research problem, then explains the followed steps in a distinct way.
* Methodology part of the paper is well organized with information on data, data pre-processing and statistical analysis.
* Authors share how BMD assessments are conducted and cite their associated references where they use AI algorithms to estimate the vertebral bone mineral density using CT data of the chest organs, which allows for precise measurement of the BMI of individuals aged 50 and older.
* The study also makes use of local data on the residents of individual districts of the metropolitan area, taking into account individual spatial information. This helps in decision-making in the field of social and health care of the residents of a municipality.
* Considering this study is a part of a pilot project, it provides insights into the practicality of the project.

Weaknesses:
* Analysis includes a unbalanced sample size, which were 1135 women and 718 men aged 50 years and older from only three medical organizations in Moscow. This could limit the generalizability of the study's findings to the wider population of Moscow residents and may result in biased decision making.
* The study's focus on BMD as the only measure of living conditions could limit the study's insights into other factors that may affect the health outcomes of residents.
* The study also does not provide a detailed discussion of the type of AI algorithm used to estimate the vertebral bone mineral density.
* The organization paper exceeds the 2 page limit criteria.
* The paper concludes with contribution of the study, excludes limitations and future work of the study.


**Suggested Changes:**

Suggested Changes:
* The paper mentions the use of socio-economic and medical data to analyze the urban health system, but it does not discuss any potential confounding variables or limitations that could affect the results. A more thorough discussion of these factors could help to contextualize the findings and highlight areas for future research.
* Including more context and background information could help readers to better understand the significance and implications of the study.
* In the methodology part, paper jumps between different sections and topics without a clear structure or flow. Reorganizing the paper and providing clear subheadings could make it easier for readers to follow the argument and understand the key points. For example, separating data part from methodology and creating a data heading specifically for data collection, anonymization and pre-processing part.
* Considering the page limit for this submission, adding 1 or 2 sentences about shortcomings and future work of the paper could help other researchers.

---

### Official Review · Reviewer_H4Qn · 2023-04-02

**Confidence:** 2

**Summary Of Contributions:**

The aim of the study is to determine the differences in the distribution of BMD indicators (vertebral bodies) according to CT data of the chest organs. Authors aim to attract the attention of urban designers and politicians with regards to sustainability and the need to analyze information correctly.

**Rating:**

Needs Clarification (NC): a submission which does not meet the reviewing criteria and needs clarification for its described problem or solution

**Strengths And Weaknesses:**

Strengths:

- Data anonymization section is good as the data collected consists of patient information.
- Statistical analysis sections involves using relevant packages.

Weaknesses:

- Aim of the study mentioned in the abstract is not clear. Sentences are too long to read. This is observed in multiple sections. Sentences can be broken down into shorter versions making it easier for reader to understand. This makes findings very hard to understand.
- It is not clear what data format was used - whether it was images or numerical or both. That makes it difficult to reproduce the results.
- Author contributions and acknowledgments section seems incomplete. Hence it seems that formatting is yet to be done.

**Suggested Changes:**

- How is this study related to learning representations? It would be good to add some sentences that describe the connection between the study and ICLR conference theme.
- While it is important to attract the attention of urban designers and politicians about certain issues - it is also important to communicate findings effectively. The sentences are too long to read. They can be broken down further.
- Results section includes the phrase - AI calculations. It would be good to rephrase the same or define what AI calculations mean.

---

### Official Review · Reviewer_XCPw · 2023-04-04

**Confidence:** 4

**Summary Of Contributions:**

This study investigates the comparability and relevance between scientific studies, taking into account important features of urbanicity (degree of urbanness), geospatial information and socio-economic properties of studied communities. It utilises an AI algorithm and conducts hypothesis testing to analyse the musculoskeletal health of older Moscow residents during COVID-19 (2021).

**Rating:**

Great Start (GS): a submission which meets some of the reviewing criteria but has room for improvement

**Strengths And Weaknesses:**

### Quality:
(+) The authors adhere to scientific practices to ensure the rigour of their study, from data collection, to data anonymization, to data analysis. They use AI as a tool to perform BMD assessment, then do hypothesis testing to understand the significance and plausibility of their results.
(-) The authors verify the hypothesis that territorial attributes are important for data analysis and understanding. However, it is unclear what these specific attributes are, how/why they impact the data and how researchers could reasonably analyse this data.

### Clarity:
(-) The writing could be further improved via more direct sentence structures and unambiguous word choices. For example, could the authors clarify what is meant by an "opportunistic analysis", "individual spatial information", "interspatial peculiarities"?

### Significance:
(+) This study makes a broader point that data and conclusions may not be directly comparable across geospatial regions (e.g. Moscow vs. the US) due to confounding factors of urbanicity, socio-economic differences, environmental factors.
(+) This study has wider implications on how the design and policy choices of urban planners and governmental officials could impact the long-term health of occupants. The authors conclude the report with a call for sustainable development.

**Suggested Changes:**

1. The authors may consider providing technical details about the AI algorithm and data used in this study, to promote reproducibility and soundness. For instance, which pre-trained neural networks were chosen and what was the rationale behind this choice? I understand that the data might be sensitive and ill-suited for open-sourcing but could the authors elaborate on the criteria for data collection and samping?
2. The first aim of this study is the "identification of interspatial peculiarities of living in a particular area by estimating BMI in Moscow residents aged 50 and older". Could the authors further elaborate on their conclusions, by explaining what "interspatial peculiarities" they have discovered?
3. The authors present evidence that territorial attributes influence the conclusions one could draw from data. It would also be informative to hear the authors' thoughts on best practices in research, in terms of how to compare across, gather insights from and build upon scientific studies conducted in different geo-political contexts.

---

### Comment · Area_Chair_AKtY · 2023-06-06
**archivial**

 This work does not meet the threshold for archival and does not contain the URM statement

---

### Meta-Review · Area_Chair_AKtY · 2023-04-03

**Recommendation:** Invite to revise
**Confidence:** 4

**Metareview:**

From the reviews, I collect the following impressions:
- Clarity: some parts of the paper provide interesting information, but in general, the writing can be improved, especially to shape the study purpose and the method involved. I also notice that the Appendix seems to be quite incomplete
- Correctness: no major concerns are raised about correctness while missing methodological information (e.g., details about "AI algorithm" used in the study, just barely mentioned) makes it challenging to analyze the correctness
- Reproducibility: the missing information harms the paper's reproducibility, which seems difficult to achieve at the moment, even from a high-level perspective.
- Basic requirements: the paper exceeds the two-page limit. References also exceed the text boundaries.

Overall, the reviewers seem to have a good feeling about the potential impact of this study, especially due to its relation with relevant social problems. However, the paper's shortcomings are significant, putting it below the bar for Clarity, Correctness, and Reproducibility requirements.

**Summary:**

The paper aims to study bone mineral density in Moscow area in people older then 50. The paper poses an interesting problem, but the overal paper seems not yet ready and might benefit further revision on misisng details and clarity.

**Comments And Feedback To The Authors:**

The authors point toward an exciting direction, but the work can obtain the correct positioning and attention only after carefully revising the presentation. The two reviewers pointed to all the significant aspects I suggest addressing. Even if the clinical trial link contains several useful pieces of information, making the paper more self-contained is essential for the reader and makes the research more accessible. I would also suggest shaping the introduction more softly: introducing the study's goal and its relevance, highlighting the problems to face, and proposing the paper's solution. Also, some figures for the methodology and results wrapped in tables might help to be more concise and clear in the exposition.

**Reason For Not Giving A Higher Recommendation:**

The paper does not meet Clarity, Correctness, and Reproducibility requirements, and also exceeds the two-pages limit. Modifications require major revisions of writing and presentation.

**Reason For Not Giving A Lower Recommendation:**

N/A

---

### Decision · Program_Chairs · 2023-04-08

No revision received; not invited to archive